# FEASIBLE ALGORITHMIC RECOURSE WITHOUT EXPLICIT STRUCTURE PRIOR

## ABSTRACT

In order to ensure that vulnerable end-users have a clear understanding of decisions made by black-box models, algorithmic recourse has made significant progress by identifying small perturbations in input features that can alter predictions. However, the generated counterfactual examples in real-world scenarios are only feasible and actionable for end-users if they preserve the realistic constraints among input features. Previous works have highlighted the importance of incorporating causality into algorithmic recourse to capture these constraints as causal relationships. Existing methods often rely on inaccessible prior Structural Causal Models (SCMs) or complete causal graphs. To maintain the causal relationships without such prior knowledge, we contribute a novel formulation that exploits the equivalence between feature perturbation and exogenous noise perturbation. To be specific, our formulation identifies and constrains the variation of exogenous noise by leveraging recent advancements in non-linear Independent Component Analysis (ICA). Based on this idea, we introduce two instantiated methods: Algorithmic Recourse with L2 norm (AR-L2) and Algorithmic Recourse with Nuclear norm (AR-Nuc). Experimental results on synthetic, semi-synthetic, and real-world data demonstrate the effectiveness of our proposed methods.

## 1 INTRODUCTION

The abundance of big data creates opportunities to enhance decision-making in areas like finance, employment, and healthcare. Machine learning models are widely used in these domains, but its important to explain their complex decisions and safeguard the rights of end-users (Pawelczyk et al., 2020; Karimi et al., 2020). Algorithmic recourse, which modifies input features to change model predictions, have gained popularity in recent years (Wachter et al., 2017). For instance, a bank could use algorithmic recourse to inform loan applicants of actions that would lead to approval. Serving as a popular explanation tool, algorithmic recourse balance model accuracy and explainability (Pawelczyk et al., 2020; Karimi et al., 2020; Mahajan et al., 2019; Kanamori et al., 2021).

One fundamental challenge for algorithmic recourse is to generate feasible real-world examples (Karimi et al., 2020). To be specific, feasibility refers to preserving realistic constraints among input features. Despite providing insight into black-box ML models, current algorithmic recourse often fails to offer actionable recommendations for individual users. For instance, suggesting to increase in education level and a decrease in age for loan approval is meaningless to end-users, as shown in Figure 1(b).

Researchers have explored realistic constraints in generating examples, considering causality (Mahajan et al., 2019; Pawelczyk et al., 2020; Kanamori et al., 2021). From the perspective of the Structural Causal Model (SCM) (Pearl, 2009), achieving feasibility is equivalent to preserving the structural causal relationships among input features. For example, methods using the distance between perturbed and source-determined features regulate explanation generation when the underlying SCM is known (Mahajan et al., 2019). When the SCM is unknown but the causal graph is available, (Karimi et al., 2020) proposes to approximate SCMs using the Gaussian process or CVAE. However, either the full SCM or the causal graph knowledge is often limited in realistic cases.

In this paper, we aim to provide feasible and interpretable algorithmic recourse without relying on SCMs or causal graphs in prior. To this end, previous claims suggest preserving linear/nonlinear structural functions among input features (Mahajan et al., 2019). However, the lack of prior causal

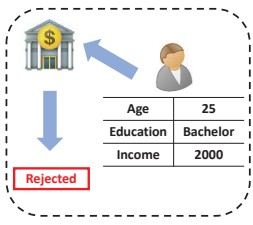
(a) Factual example.

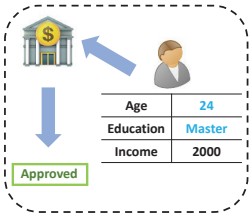
(b) Vanilla recourse.

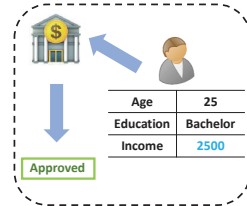
(c) Actionable recourse.

**Figure 1: Distinctions among factual examples rejected by the ML decision model, vanilla algorithmic recourse that alter predictions but are not actionable, and feasible algorithmic recourse complying with realistic causal relationships.**

knowledge makes direct identification or approximation of these functions impossible (Shimizu, 2014; Hyvarinen et al., 2019; Karimi et al., 2020). To overcome this, we propose that the process of the algorithmic recourse on the input features can be modeled as solely the manipulation of the exogenous noise of each sample, while the structural causal relationships among features remain. To the best of our knowledge, we are the first to propose this idea. To instantiate this idea, we indirectly preserve causal relationships by identifying and constraining the variation of exogenous noise in aid of the non-linear Independent Component Analysis (ICA) (Hyvarinen et al., 2019; Hyvarinen & Morioka, 2017). Theoretically, we show that exogenous noise can be identified in a reliable manner by constructing an exogenous regressor. Subsequently, we further prove that the variation of the exogenous noise is governed by that of representations learned by the exogenous regressor under mild conditions. Practically, we propose two practical methods, AR-$\mathcal{L}$2 and AR-Nuc, which constrain the magnitude and sparsity of variations in exogenous representations, respectively. Extensive experimental results verify that our methods: (a) significantly improve the preservation of causal relationships for algorithmic recourse; (b) successfully achieve the alteration of predictions with little cost.

## 1.1 RELATED WORK

**Algorithmic recourse**  Traditional local explanation methods for black-box ML models on data, such as tabular data, are crucial for users to interpret decisions (Ribeiro et al., 2016). However, these explanations often differ from complex ML models. Algorithmic recourse (or counterfactual explanation) offers consistent and interpretable examples as an alternative (Wachter et al., 2017; Mahajan et al., 2019; Karimi et al., 2020). They can be categorized into gradient-based methods (Wachter et al., 2017; Moore et al., 2019; Mahajan et al., 2019; Karimi et al., 2020) and linear programming (Ustun et al., 2019; Kanamori et al., 2021). Recent discussions have also addressed fairness, transportability, and reliability issues in algorithmic recourse (Black et al., 2021; von Kügelgen et al., 2022).

While the definition of algorithmic recourse has parallels with adversarial examples (Brown et al., 2017; Santurkar et al., 2021), the **biggest** distinction between the two directions is that the former only aims to explain to the model while the latter aims to suggest both interpretable and actionable recommendations to end-users (Karimi et al., 2021a). Unfortunately, most of the current algorithmic recourse methods lack such capability by ignoring the relationships and constraints among the input features (Ustun et al., 2019; Karimi et al., 2020).

**Causality for feasible algorithmic recourse**  To overcome the above-mentioned challenge, several recent works have suggested the incorporation of causality into the algorithmic recourse (Mahajan et al., 2019; Karimi et al., 2020; Kanamori et al., 2021; Ustun et al., 2019). From the view of causality, the infeasibility of vanilla algorithmic recourse stems from the fact that such recourse are generated by independently manipulating each input feature. As a consequence, the causal relationship/constraints are broken during the generation process (as shown in Figure 1(b)). Accessing the entire SCM model, (Mahajan et al., 2019) suggests regularization of algorithmic recourse by minimizing differences between perturbed features and their parent-generated counterparts. Building on this concept, several works(von Kügelgen et al., 2022; Karimi et al., 2020; 2021b) seek optimal intervention feature sets with minimal cost. These methods relax the requirement from knowing

the whole SCM model to knowing the causal graph. For instance,(Karimi et al., 2020) proposes approximating the SCM model using Gaussian process or Conditional Variational Encoder(CVAE). Moreover, (Kanamori et al., 2021) explores different cost setups with accessible causal graphs.

## 2 BACKGROUND

**Notations**   We assume a binary classifier $h$ as the underlying ML model (Kanamori et al., 2021; Karimi et al., 2020; Mahajan et al., 2019), while our method also allows for a multi-categorical classifier. We use a dataset $\mathcal{D}$ with $M$ samples, each consisting of $n$ input characteristics $\mathbf{x}$ and output labels $\mathbf{y}$. We index data points as $1 \le i \le M$ and features as $1 \le j \le n$. The goal of algorithmic recourse is to answer why some individuals were denied loans and what actions they could take to increase approval chances. We search for the closest algorithmic recourse $\mathbf{x}^{\mathrm{R}}$ for a given factual sample $\mathbf{x}^F$ using model $\mathcal{M}$: $\mathbf{x}^{\mathrm{R}} \in \arg\min_{\mathbf{x} \in \mathcal{X}} \mathrm{d}\left(\mathbf{x}, \mathbf{x}^{\mathrm{F}}\right)$ s.t. $h(\mathrm{x}) \ne \mathbf{y}$, where $\mathrm{d}$ refers to some pre-defined distance functions. To be specific, the counterfactual $\mathbf{x}^{\mathrm{R}}$ is often computed as $\mathbf{x}^{\mathrm{R}} = \mathbf{x}^{\mathrm{F}} + \delta$ by adding some perturbations $\delta$ on $\mathbf{x}^{\mathrm{F}}$.

**Structural Causal Models.**   From the perspective of causality, it is crucial to first identify the generation of the observational data (Pearl, 2009). Specifically, the data-generating process of X is described by a Markov structural causal model (SCM) $\mathcal{V} = (X, \mathrm{F}, P_\sigma)$ describes the causal relations between $n$ features in $X = \{\mathbf{x}_1, \mathbf{x}_2, \ldots, \mathbf{x}_n\}$ as: $\mathbb{F} = \left\{\mathbf{x}_j := f_j\left(\mathbf{x}_{\mathrm{pa}(j)}, \sigma_j\right)\right\}_{j=1}^{n}$ (Pearl, 2009), where $P_\sigma = P_{\sigma_1} \times \ldots \times P_{\sigma_n}$ and $\mathbb{F}$ is the set of assignment functions $f_j$ which maps feature $\mathbf{x}_j$ to its causal parents $\mathbf{x}_{\mathrm{pa}(j)}$. Following previous protocols (Karimi et al., 2020; Mahajan et al., 2019), we here assume the non-existence of unmeasured confounders, i.e., the *causal sufficiency* assumption (Pearl, 2009), such that the exogenous noise $\sigma_j$ are mutually independent. Meanwhile, the SCM is often coupled with its intuitive illustration, i.e., the causal graph $G$, which is formed by a one-to-one mapping from each variable $x_j$ to a node in $G$ and directed edges drawn from $x_j$ to $\mathbf{x}_{\mathrm{pa}(j)}$ for $j \in [n]$. To ensure the SCM is non-recursive, we follow (Karimi et al., 2020) and assume throughout that $G$ is *acyclic*.

**Interventions with Minimal Cost.**   Assuming the SCM model $\mathcal{V}$ is accessible with invertible forms, e.g., additive SCMs, (Karimi et al., 2021b) formulate the above algorithmic recourse problem as finding the optimal intervention strategy with minimal cost:

$$\mathbf{A}^* \in \underset{\mathrm{A} \in \mathcal{V}}{\arg\min}\, \mathrm{cost}\left(\mathbf{A}; x^{\mathrm{F}}\right) \quad \text{s.t.} \quad x^{\mathrm{SCF}} = \mathbb{F}_{\mathrm{A}}\left(\mathbb{F}^{-1}\left(x^{\mathrm{F}}\right)\right),\ h\left(x^{\mathrm{SCF}}\right) \ne h\left(x^{\mathrm{F}}\right) \tag{1}$$

where $\mathbf{A}^*$ directly specifies the set of feasible actions to be performed for minimally costly recourse. By the three steps of structural counterfactuals (Pearl, 2009), the counterfactual examples, i.e., $x^{\mathrm{SCF}}$, is generated based on the evidence $X^F$ and $\mathcal{V}$, and we use $\mathbb{F}_{\mathrm{A}}\mathbb{F}^{-1}$ to denote such procedure (Karimi et al., 2021b). Based on this foundation, some relaxation has been proposed in (Karimi et al., 2020) by assuming only the access of causal graph $G$ rather than the SCM $\mathcal{V}$. Unfortunately, as the ultimate goal of causal discovery (Zhu et al., 2019), the prior causal graph still restricts the application of algorithmic recourse. Hence, how to maintain the causal relationship without prior knowledge is of urgent need in many scenarios.

**Connections between ICA and Causal Inference.**   ICA aims to identify *mutually independent* source signal $S$ from mixed observations $T$ via a mixture function $\tilde{f}$: $T = \tilde{f}(S)$, e.g., the cocktail party problem (Haykin & Chen, 2005). While the traditional ICA theory can successfully identify the non-Gaussian distributions of $\tilde{f}$ and $\sigma$ under the condition that $\tilde{f}$ is a linear function (Shimizu, 2014), practical applications often involve non-linear functions for $\tilde{f}$, making it extremely challenging to directly infer $\tilde{f}$ without additional information (Hyvarinen & Morioka, 2017). Previous advances in causal inference have utilized ICA to identify different forms of SCMs (Shimizu, 2014; Gresele et al., 2021). For instance, the identification of causal graph $G$ for linear and additive SCMs, i.e., $\mathbf{x}_j := w_j\mathbf{x}_{\mathrm{pa}(j)} + b_j$ for $j \in [n]$ ($w, b$ are linear coefficients), can be reformulated into a linear ICA problem (Zhu et al., 2019; Shimizu, 2014). On the other side, (Gresele et al., 2021) has contributed novel identification results for non-linear ICA based on independent causal mechanisms of source signals. These works build up the foundation of our work to identify and constrain the exogenous noise for learning actionable algorithmic recourse which is the goal of this paper.

## 3 Methods

### 3.1 Identification of Variation of Exogenous Noise

**Relating Algorithmic Recourse to ICA.** Throughout our paper, we clarify again that all our derivations and designs are based on the assumptions that the factual examples, i.e., $X^F$, are generated based on the additive, non-recursive, and Markov SCMs described in Section 2. In other words, we model the data generation using additive SCMs $\mathcal{V}$ with causal sufficiency assumption and acyclic causal graph $G$. To generate actionable counterfactual examples $\mathbf{x}^R$ without $G$ and $\mathcal{V}$, we reformulate the generation process as the modifications of exogenous noise. More formally, we depict the generation of factual data $\mathbf{x}^F$ and recoursed $\mathbf{x}^R$ in the form of additive SCMs with structural functions $f : \mathbb{R}^n \mapsto \mathbb{R}^n$ and the exogenous noise $\sigma^F \in \mathbb{R}^n$:

$$\mathbf{x}^{\mathrm{F}} = f(\mathbf{x}^{\mathrm{F}}) + \sigma^F \Rightarrow \mathbf{x}^{\mathrm{R}} = f(\mathbf{x}^{\mathrm{R}}) + \sigma^R, \tag{2}$$

where $\sigma^R$ is the results of modifications on $\sigma^F$: $\sigma^R = \sigma^F + \delta$ ($\delta \in \mathbb{R}^n$ refers to the variation of the exogenous noise). Notably, the above formulation is just an aggregated variant of separate formulation of additive SCM for each node, i.e., $x_j := f_j\left(\mathbf{x}_{\mathrm{pa}(j)}\right) + \sigma_j, j \in [n]$ (We have provided more detailed illustration on this in the appendix A.1 for saving space). Intuitively, we model the process of algorithmic recourse as the solely manipulation of the exogenous noise, while the structural equations $f$ which characterize the causal relationships are kept invariant.

**Identification of the exogenous noise.** Assuming that the operator $g = (I - f)^{-1}$ is invertible ($I$ is the identity mapping), we turn equation equation 2 into $\mathbf{x}^{\mathrm{R}} = g(\sigma^{\mathrm{R}})$. As we have assumed the Markov SCMs, the exogenous noise elements $\sigma$ are mutually independent (Pearl, 2009). Consequently, the reformulation $\mathbf{x}^{\mathrm{R}} = g(\sigma^{\mathrm{R}})$ has natural connections to the formulation of ICA (Hyvarinen & Morioka, 2017), i.e., $T = f(S)$, where the signal $\sigma^{\mathrm{R}}$ can be interpreted as the source signal $S$, the function $g$ represents the mixing process $\tilde{f}$, and $\mathbf{x}^{\mathrm{R}}$ represents the mixed observations $T$.

Based on above observations, we determine the variability value of $\sigma^{\mathrm{R}}$ using advanced non-linear ICA techniques (Hyvarinen et al., 2019). By introducing an observed auxiliary variable $c$, we ensure that each element $\sigma_{j1}$ depends statistically on $\mathbf{c}$ but is conditionally independent of the other elements $\sigma_{j2}$ (Hyvarinen et al., 2019) [1]. We randomly permute the sample order of $\mathbf{y}$ to eliminate correlations with $\sigma$ and construct the modified data $\mathcal{D}^A$ using permuted $\mathbf{y}$. Finally, we employ a discriminative model called the **exogenous regressor** to distinguish between $\mathcal{D}^A$ and $\mathcal{D}$ through a non-linear regression system as follows:

$$\min_{\theta} \sum_{i=1}^{M} l\left(\frac{1}{1 + \exp(-r(\mathbf{x}_i, \mathbf{y}_i))}, o_i\right) \quad s.t. \quad r(\mathbf{x}, \mathbf{y}) = \sum_{j=1}^{n} \psi_j^{\theta}\left(\phi_j^{\theta}(\mathbf{x}), \mathbf{y}\right), \tag{3}$$

where the binary labels $\mathbf{o}$ indicate the source of the data as either $\mathcal{D}$ or $\mathcal{D}^A$. The functions $\psi_j^{\theta} : \mathbb{R}^2 \mapsto \mathbb{R}$ and $\phi_j^{\theta} : \mathbb{R}^n \mapsto \mathbb{R}$ are non-linear representations parameterized by $\theta$, implemented using deep networks (Hyvarinen et al., 2019). In this paper, we refer to $\phi$ as the "exogenous representations" for simplicity. We then offer theoretical insights into the behavior of the learned $\phi_{\theta}(\mathbf{x})$ to support the validity of our exogenous regressor as follows:

**Theorem 3.1** (Identification of $\sigma$). *Assume:*

*(a) The exogenous noise $\sigma$ is conditionally exponential of order $K$ of $\mathbf{y}$. Consequently, the conditional probability density of $\sigma$ given $\mathbf{y}$ can be written for each feature $1 \le j \le n$:*

$$p\left(\sigma_j \mid \mathbf{y}\right) = \frac{Q_j\left(\sigma_j\right)}{Z_j(\mathbf{y})} \exp\left[\sum_{k=1}^{K} \tilde{q}_{jk}\left(\sigma_j\right) \lambda_{jk}(\mathbf{y})\right], \tag{4}$$

*where $Q_j, Z_j, \tilde{q}_{jk}$ and $\lambda_{jk}$ as scalar-valued functions. Meanwhile, for each $j$, the sufficient statistics $\tilde{q}_{jk}$ are assumed linearly independent over $k$.*

---

[1]Notably, such auxiliary variables could be historical observations, the time variables or the class labels (Hyvarinen et al., 2019). The class label $\mathbf{y}$ serves as a suitable choice for $c$ (Hyvarinen & Morioka, 2017).

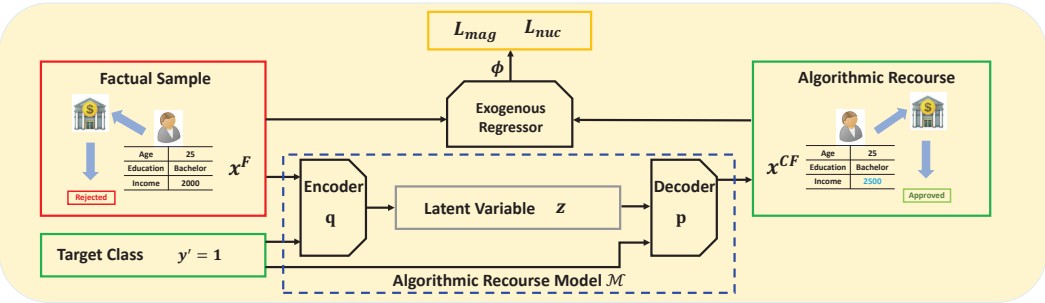

**Figure 2: Framework of our method.**

(b) *There exists $nk + 1$ realizations of $\mathbf{y}$ as $\{\mathbf{y}\}_{l=0}^{nk}$ such that the matrix with size $nk \times nk$:*

$$\mathbf{L} = \left( \begin{array}{c} \lambda_{11}(\mathbf{y}_1) - \lambda_{11}(\mathbf{y}_0), \ldots, \lambda_{11}(\mathbf{y}_{nk}) - \lambda_{11}(\mathbf{y}_0) \\ \vdots \\ \lambda_{nk}(\mathbf{y}_1) - \lambda_{nk}(\mathbf{y}_0), \ldots, \lambda_{nk}(\mathbf{y}_{nk}) - \lambda_{nk}(\mathbf{y}_0) \end{array} \right) \tag{5}$$

*is invertible.*

(c) *The trained (deep) logistic regression system in equation 3 has the universal approximation capability to distinguish $\mathcal{D}$ from $\mathcal{D}^A$.*

*Then, in the case of infinite samples, the representations $\phi^\theta(\mathrm{x})$ identifies $\sigma$ up to a linear transformation of point-wise statistics $\tilde{\mathbf{q}}$:*

$$\tilde{\mathbf{q}}(\sigma) = \mathbf{A}\phi_\theta(\mathrm{x}) + \mathbf{b}, \tag{6}$$

*where $\mathbf{A}$ and $\mathbf{b}$ are fixed but unknown matrices.*

Notably, although the above theorem provides the general case for any $k \geq 1$, we will only treat the cases when $k = 1$ throughout the following parts.

**Constraining the variation of the exogenous noise.** Based on the obtained results, we establish a connection between the variation of exogenous representations $\phi$ and exogenous noise $\sigma$ as follows:

$$\mathbf{A}\left(\phi(\mathbf{x}^F) - \phi(\mathbf{x}^{AR})\right) = \tilde{\mathbf{q}}(\sigma^F) - \tilde{\mathbf{q}}(\sigma^R). \tag{7}$$

Building on previous analyses, we aim to limit the variation of exogenous noise as $\sigma^F - \sigma^R$. This is based on the intuition that excessive variation in $\sigma^F - \sigma^R$ diminishes the influence of structural functions $f$. Essentially, significant variability in $\sigma$ closely resembles point-wise intervention. Initially, we assume the variation in exogenous representation for a batch of samples, denoted as $\mathbf{H}$, as follows:

$$\mathbf{H} = \{\phi(\mathbf{x}_1^F) - \phi(\mathbf{x}_1^{AR}), \phi(\mathbf{x}_2^F) - \phi(\mathbf{x}_2^{AR}), \ldots, \phi(\mathbf{x}_{M_b}^F) - \phi(\mathbf{x}_{M_b}^{AR})\}. \tag{8}$$

Consequently, we will further demonstrate that constraining the sparsity and magnitude of $\mathbf{H}$ adequately restricts the corresponding characteristics of $\sigma^F - \sigma^R$, respectively.

**Sparsity Constraint** The intuition behind the sparsity constraint is that we expect the number of perturbed input features to be small. To this end, we restrict the variation vector of the exogenous noise to be sparse by minimizing the rank of the matrix $\mathbf{H}$. As the rank of a matrix is well approximated by its nuclear norm, we propose the algorithmic recourse with Nuclear Norm (AR-Nuc) by optimizing the nuclear norm of $\mathbf{H}$ as $\|\mathbf{H}\|_*$: $min_{\mathbf{X^{AR}}} : \mathcal{L}_{\mathrm{nuc}} = \|\mathbf{H}\|_*$.

The optimization process only updates the generated $\mathbf{X^{AR}}$ while keeping $\phi$ and factual input $\mathbf{x}^F$ fixed. However, one might be confused on how the sparsity of $\mathbf{H}$ correlates to that of $\mathbf{H}^0 = \{\sigma_1^F - \sigma_1^{AR}, \sigma_2^F - \sigma_2^{AR}, \ldots, \sigma_{M_b}^F - \sigma_{M_b}^{AR}\}$. To answer this question, we provide theoretical insights to bridge the sparsity of $\mathbf{H}^\sigma$ to that of $\mathbf{H}$ as follows:

**Theorem 3.2** (Connection between $\mathbf{H}$ and $\mathbf{H}^0$). *Assume:*

(a) *The sufficient statistics $\tilde{\mathbf{q}}_{ij}$ are differentiable almost everywhere, and for fixed $i$, $\{\tilde{\mathbf{q}}_{ij}\}_{j=1}^k$ are linearly independent on any subset of $\mathbb{R}^n$ with measure greater than zero.*

(b) *For fixed $i$, the representation functions $\phi_{ij}$ are linearly independent on any subset of $\mathbb{R}^n$ with measure greater than zero.*

Then, for $k = 1$, the resulting matrix $\mathbf{A}$ is of full-rank and invertible. Meanwhile, the sparsity of $\mathbf{H}^0$ is governed by that of $\mathbf{H}$.

**Magnitude Constraint** Beyond the sparsity constraint, we restrict the exogenous noise to vary in a small magnitude as well. To this end, we design another method named the algorithmic recourse with $\mathcal{L}2$ norm (AR-$\mathcal{L}2$). More specifically, we first provide the following theorem to argue that optimizing the $\mathcal{L}2$ norm of the variation of exogenous representations, $\phi(\mathbf{x}^F) - \phi(\mathbf{x}^{AR})$, is enough to constrain that of $\sigma^F - \sigma^R$:

**Theorem 3.3.** *Assume the sufficient statistics $\tilde{\mathbf{q}}$ (k=1) is a bi-Lipschitz function of $\sigma$, then $\|\sigma^F - \sigma^R\|_2$ is governed by $\|\phi(\mathbf{x}^F) - \phi(\mathbf{x}^{AR})\|_2$, where $\|\cdot\|_2$ is the $\mathcal{L}2$ norm.*

Therefore, we propose the algorithmic recourse with $\mathcal{L}2$ Norm (AR-$\mathcal{L}2$) by optimizing the $\mathcal{L}2$ norm of $\mathbf{H}$ on a batch of samples: $min_{\mathbf{X}^{AR}} : \mathcal{L}_{\text{mag}} = \|\mathbf{H}\|_2$. To save space, all proofs are placed in the appendix A.3.

## 3.2 CHOICE OF ALGORITHMIC RECOURSE MODEL

As the conditional variational autoencoder (CVAE) provides a flexible and reliable approach (Mahajan et al., 2019; Pawelczyk et al., 2020), we adopt the previous proposed CFVAE model (Mahajan et al., 2019) to generate the algorithmic recourse with altered prediction and minimal cost in this paper. More specifically, we achieve this by maximizing the log-likelihood of $P(\mathbf{x}^{AR} \mid \mathbf{x}^F, \mathbf{y}')$, where $\mathbf{y}'$ refers to the target prediction altered from the original decision $\mathbf{y}$. Following previous protocol(Mahajan et al., 2019), we instead maximize the evidence lower bound (ELBO) of $P(\mathbf{x}^{AR} \mid \mathbf{x}^F, \mathbf{y}')$ by following:

$$\mathbb{E}_{q(\mathbf{z}|\mathbf{x}^F,\mathbf{y}')} \log p\left(\mathbf{x}^{AR} \mid \mathbf{z}, \mathbf{y}', \mathbf{x}^F\right) - \mathcal{KL}\left(q\left(\mathbf{z} \mid \mathbf{x}^F, \mathbf{y}'\right) \| p\left(\mathbf{z} \mid \mathbf{y}', \mathbf{x}^F\right)\right). \tag{9}$$

where we first arrive the latent representations $\mathbf{z}$ via the encoder $q(\mathbf{z} \mid \mathbf{x}^F, \mathbf{y}')$ and then generate the counterfactual $\mathbf{x}^{AR}$ via the decoder $p\left(\mathbf{x}^{AR} \mid \mathbf{z}, \mathbf{y}', \mathbf{x}^F\right)$. Meanwhile, the prior conditional density of $\mathbf{z}$ is sampled from a normal distribution: $p(\mathbf{z} \mid \mathbf{y}', \mathbf{x}^F) \sim N(\mu_{y'}, \sigma_{y'}^2)$ to achieve a closed form of the KL-divergence. For realizations, we adopt the $\mathcal{L}1$ norm to measure the reconstruction loss, with an additional Hinge loss to force the ML model $h$ to alter the prediction from $\mathbf{y}$ to $\mathbf{y}'$:

$$\begin{cases} \mathcal{L}_{\text{recon}}(\mathbf{x}^F, \mathbf{x}^{AR}) = \log p\left(\mathbf{x}^{AR} \mid \mathbf{z}, \mathbf{y}', \mathbf{x}^F\right) = \|\mathbf{x}^{AR} - \mathbf{x}^F\|_1, \\ \mathcal{L}_{\text{hinge}}(h(\mathbf{x}^{AR}), \mathbf{y}', \beta) = \max(h_{\mathbf{y}}(\mathbf{x}^{AR}) - h_{\mathbf{y}'}(\mathbf{x}^{AR}), -\beta), \end{cases} \tag{10}$$

where $h_{\mathbf{y}}(\mathbf{x}^{AR})$ refers to the predicted score (e.g., a probability in $[0, 1]$) from $h$ at class $\mathbf{y}$, $\beta$ is the hyper-parameter to control the margin. Finally, by performing the monto-carlo approximation and sampling from the encoder $q(\mathbf{z} \mid \mathbf{x}^F, \mathbf{y}')$, we express the original loss for optimizing $\mathcal{M}$ on a batch sample with size $M_b$ as follows:

$$\mathcal{L}_{\text{ori}} = \sum_{i=1}^{M_b} \mathcal{L}_{\text{recon}}(\mathbf{x}_i^F, \mathbf{x}_i^{AR}) + \mathcal{L}_{\text{hinge}}(h(\mathbf{x}_i^{AR}), \mathbf{y}_i', \beta) + \mathcal{KL}\left(\mathbf{y}_i', \mathbf{z}_i, \mathbf{x}_i^F\right), \tag{11}$$

where $\mathcal{KL}\left(\mathbf{y}_i', \mathbf{z}_i, \mathbf{x}_i^F\right)$ refers to the empirical estimation of $\mathcal{KL}\left(q\left(\mathbf{z} \mid \mathbf{x}^F, \mathbf{y}'\right) \| p\left(\mathbf{z} \mid \mathbf{y}', \mathbf{x}^F\right)\right)$.

The loss $\mathcal{L}_{\text{nuc}}$ and $\mathcal{L}_{\text{mag}}$ can be incorporated into the above objective to preserve the causal relationships. Therefore, the overall objective function can be written as $\mathcal{L}_{\text{ori}} + \alpha_{nuc}\mathcal{L}_{\text{nuc}}$ and $\mathcal{L}_{\text{ori}} + \alpha_{mag}\mathcal{L}_{\text{mag}}$, where $\alpha_{mag}$ and $\alpha_{nuc}$ are hyper-parameters to balance the original counterfactual generation and our causal regularization. The pseudo-code of the whole algorithm can be found in the appendix (see A.2).

## 4 EXPERIMENTS

In this section, we first introduce the baselines we compared, together with the evaluation metrics. Then we provide experimental results on a synthetic dataset, a semi-synthetic dataset, and a real-world dataset.

**Baselines and implementations**    Overall, our baselines can be divided into three levels: (a) Vanilla algorithmic recourse methods without any prior knowledge. Such methods include (1) the CFVAE model we introduced before (Mahajan et al., 2019) and (2) the CEM model, which models the perturbation using the auto-encoder structure (Dhurandhar et al., 2018); (b) Partial oracle baselines with the prior causal graph. In detail, we choose the minimal-intervention framework proposed in (Karimi et al., 2020) by using CVAE to approximate the SCM model at the probabilistic level, which we call the CFA method. (c) Oracle baselines, which refers to the methods with the whole SCM model as a prior. Such a method is implemented on the basis of the CFVAE regularized by the causal distance proposed in (Mahajan et al., 2019), which we call the AR-SCM method. We implement the CFA method in two versions: CFA-All (CFA-a) and CFA-Partial (CFA-p), allowing interventions on all features or only a subset, respectively. More details are present in the appendix A.4.

**Metric**    We evaluate the quality of generated algorithmic recourse using the following metrics (Mahajan et al., 2019; Karimi et al., 2020): (1) Feasibility Score (%): Percentage of individuals whose algorithmic recourse satisfy the prior monotonic constraints, indicating feasibility; (2) Distribution Score: Log-likelihood ratio of generated algorithmic recourse compared to the given causal edges, indicating compliance with the SCM model. The distribution score w.r.t this edge equals to $\log p\left(\mathbf{x}_j^{AR} \mid \mathbf{x}_{Pa(j)}^{AR}\right) / \log p\left(\mathbf{x}_j^F \mid \mathbf{x}_{Pa(j)}^F\right)$; (3) Validity (%): Percentage of individuals with favorable predictions from algorithmic recourse; (4) Proximity: Average $\mathcal{L}$1-distance between counterfactual and original features for continuous features, and number of mismatches for categorical features. We conduct experiments and compute metrics in two settings: in-sample, testing the model on training samples, and out-of-sample, testing on samples outside the training dataset without output labels. In our experiments, we mainly answer two questions: (a) *How does our method perform on preserving the causal relationship?* (b) *Does our method sacrifice other metric (e.g., the Proximity or Validity) to improve the feasibility?*

**Synthetic dataset**    Inspired by previous protocols (Mahajan et al., 2019), we simulate a toy dataset with three features as $(\mathbf{x}_1, \mathbf{x}_2, \mathbf{x}_3)$ and one outcome variable $(\mathbf{y})$. To incorporate a monotonically increasing causal relationship between $\mathbf{x}_1, \mathbf{x}_2$ and $\mathbf{x}_3$, we adopt the following structural equations as in (Mahajan et al., 2019):

$$\mathbf{x}_1 \sim N\left(\mu_1, \sigma_1\right); \mathbf{x}_2 \sim N\left(\mu_2, \sigma_2\right); \mathbf{x}_3 \sim N\left(k_1 * \left(\mathbf{x}_1 + \mathbf{x}_2\right)^2 + b_1, \sigma_3\right);$$
$$\mathbf{y} \sim \text{Bernoulli}\left(k_2 * \left(\mathbf{x}_1 * \mathbf{x}_2\right) + b_2 - \mathbf{x}_3\right),$$

(12)

where we set $\mu_1 = \mu_2 = 50$, $\sigma_1 = 15$, $\sigma_2 = 17$, $\sigma_3 = 0.5$, $k_1 = 0.0003$, $k_2 = 0.0013$, $b_1 = b_2 = 10$ as in (Mahajan et al., 2019). Obviously, the causal relationship embodied in this dataset is $\mathbf{x}_1, \mathbf{x}_2$ increase $\Rightarrow \mathbf{x}_3$ increases; and $\mathbf{x}_1, \mathbf{x}_2$ decrease $\Rightarrow \mathbf{x}_3$ decreases. Thus the feasibility set $C$ equals to the above two constraints. For method CFA-a, we allow $\mathbf{x}_1, \mathbf{x}_2$ and $\mathbf{x}_3$ to be intervened, while only $\mathbf{x}_1$ and $\mathbf{x}_2$ are allowed to be intervened for CFA-p.

Table 1 and Figure 3 demonstrate the effectiveness of our method, AR-Nuc and AR-$\mathcal{L}$2. It achieves significant improvements in the feasibility and distribution scores. Compared to the vanilla CFVAE, our feasibility score improves by over 15%. AR-Nuc and AR-$\mathcal{L}$2 perform competitively with the ground truth approach (AR-SCM) on feasibility and distribution scores. Therefore, our methods successfully preserve the causal relationship from $\mathbf{x}_1$ and $\mathbf{x}_2$ to $\mathbf{x}_3$, and maintain validity by altering predictions with minimal cost. Notably, our methods outperform CFA-a and CFA-p, even with prior causal graph, due to better approximation of structural equations.

**German Loan Dataset**    A semi-synthetic dataset called "German Loan" was created based on the German Credit UCI dataset (Karimi et al., 2020). The dataset includes 7 variables (age, gender, education level, loan amount, duration, income, and savings) with the class label indicating loan approval. The causal graph and structural equations can be found in the appendix. For the German Loan dataset, the CFA-p method was implemented with non-interventive features (age, gender, and duration), and a constraint set (C) was used to measure feasibility, following three rules: (1) Loan amount (L) increases $\Rightarrow$ loan duration (D) increases; (2) Age (A) increases $\Rightarrow$ income (I) increases; (3) A increases $\Rightarrow$ education-level (E) increases.

As shown in Table 1 and Figure 3, our AR-Nuc and AR-$\mathcal{L}$2 outperform others in feasibility and distribution scores while maintaining 100% validity at a low proximity cost. Additionally, CFA-p

**Table 1: Results of the distribution and proximity scores on synthetic and German Loan data: Metrics are Mean±STD over 5 repeated experiments, with the best Dist_score highlighted.**

| Setting | | In-sample | | Out-of-sample | |
|---|---|---|---|---|---|
| Metric | | Proximity | D-Score | Proximity | D-Score |
| **Benchmark: Synthetic** | | | | | |
| Vanilla | CEM | 4.82±0.85 | -369.74±8.9 | 3.79±0.62 | -372.50±10.2 |
| | CFVAE | 2.12±0.51 | 2.31±0.26 | 2.09±0.55 | 2.30±0.25 |
| Partial | CFA-a | 2.24±0.07 | -4.76±2.10 | - | - |
| | CFA-p | 2.18±0.11 | -2.53±1.15 | - | - |
| Ours | AR-Nuc | 2.38±0.26 | **3.26±0.28** | 2.37±0.15 | **3.08±0.22** |
| | AR-$\mathcal{L}2$ | 2.06±0.44 | **3.03±0.12** | 2.07±0.22 | **3.12±0.05** |
| *Oracle* | AR-SCM | *2.11±0.32* | *3.58±0.21* | *2.28±0.27* | *3.66±0.08* |
| **Benchmark: German** | | | | | |
| Vanilla | CEM | 4.67±0.51 | 0.68±0.27 | 4.67±0.44 | 0.49±0.25 |
| | CFVAE | 6.14±0.13 | 1.02±0.14 | 6.15±0.15 | 1.03±0.10 |
| Partial | CFA-a | 6.04±0.20 | 0.99±0.05 | - | - |
| | CFA-p | 6.10±0.18 | 0.83±0.19 | - | - |
| Ours | AR-Nuc | 5.95±0.14 | **3.42±0.10** | 5.80±0.13 | **3.45±0.13** |
| | AR-$\mathcal{L}2$ | 6.02±0.10 | **3.35±0.08** | 6.01±0.11 | **3.40±0.07** |
| *Oracle* | AR-SCM | *6.18±0.27* | *3.49±0.17* | *6.19±0.26* | *3.51±0.09* |

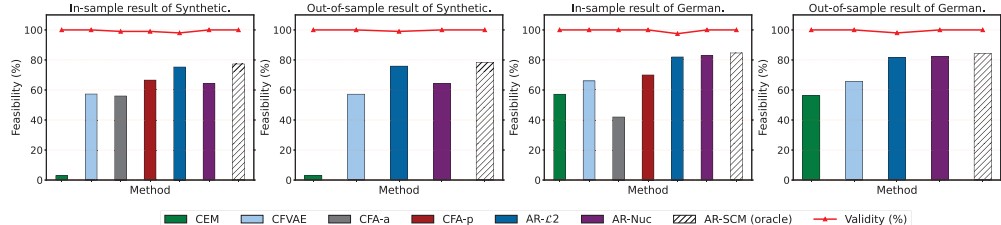

**Figure 3: Results on the feasibility and valid scores for the synthetic data and the German Loan data, where higher metric means better results.**

consistently performs better than CFA-a, highlighting that intervening on all features leads to poor feasibility performance without protection (Karimi et al., 2020). For instance, intervening on the node $I$ (income) violates the second constraint in $C$ by independently manipulating $A$ and $I$. This observation further supports our claim that modifying the exogenous noise $\sigma$ instead of point-wise intervention (Karimi et al., 2020) is more suitable for preserving causal relationships when the causal graph is inaccessible.

**Diabetes Dataset** The Diabetes dataset (Kanamori et al., 2021), collected by Smith (Smith et al., 1988), consists of medical records for Pima Indians with and without diabetes. It includes 8 input features such as pregnant status, blood pressure, skin thickness, body mass index (BMI), and age, with the class label indicating diabetic conditions. No prior SCM model or distribution score is available for Diabetes . To discover the causal structure, we use the CFA-a

**Table 2: Proximity score of the Diabetes dataset. We refer to the CFA method with all nodes allowed to be intervened as CFA-Discover, as we pre-train a causal discovery model to provide the prior causal graph.**

| Setting | | In-sample | Out-of-sample |
|---|---|---|---|
| Vanilla | CEM | 7.42±0.11 | 7.43±0.08 |
| | CFVAE | 16.49±0.52 | 16.19±0.47 |
| Partial | CFA-Discover | 6.67±0.26 | - |
| Ours | AR-Nuc | 6.43±0.18 | 6.40±0.16 |
| | AR-$\mathcal{L}2$ | 6.48±0.19 | 6.50±0.11 |

method (Karimi et al., 2020) with the NOTEARS method (Zheng et al., 2018)[2]. The CFA-p method cannot be implemented due to the absence of a prior causal structure. Based on prior knowledge(Kanamori et al., 2021; Smith et al., 1988), the constraint set $C$ includes three rules: (1) Blood Pressure → BMI, (2) Glucose → BMI, and (3) Skin thickness → BMI.

---

[2]https://github.com/xunzheng/notears

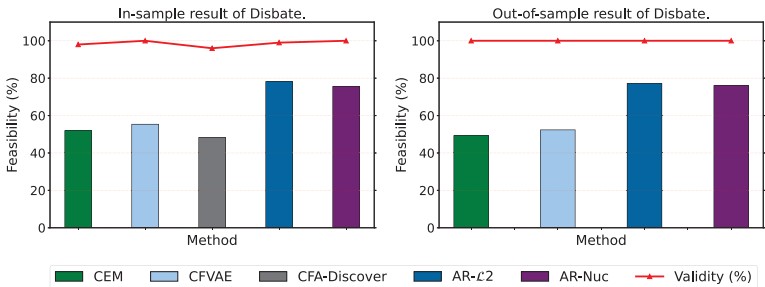

**Figure 4: Feasibility and validity scores on Diabetes .**

As shown in Table 2 and Figure 4, the insights discussed above can be similarly applied to the real-world dataset. One reason could lie in the fact that our approaches effectively constrain the variation of the exogenous noise, which further preserves the effect of the structural functions in generated examples. Meanwhile, we observe that the CFA-a method performs poorly on feasibility with no improvement compared to the vanilla CEM and CFVAE. Hence, the first-discover-then-generation approach is difficult for realistic cases, where the error of discovery and approximation will accumulate together.

**Towards High-dimensional Data**    We test the capability of our method by involving a synthetic dataset with 80 features in our study, using similar approaches as synthetic dataset (see in appendix equation A.4). The rationale behind employing synthetic data is twofold: (a) most widely used, realistic datasets possess relatively small feature dimensions; (b) real-world data with high dimensions lacks an underlying SCM, rendering it difficult to evaluate the feasibility.

**Table 3: Performance on high-dimensional data, where the time refers to the running time for recoursing per sample in the dataset.**

| Methods | | Proximity | D-Score | Validity (%) | Feasibility (%) | Time (s) |
|---|---|---|---|---|---|---|
| Vanailla | CEM | 5.61±1.24 | <-500 | 99 | <10 | - |
| | CFVAE | 2.68±0.88 | 1.52±0.36 | 100 | 35.6 | 0.12 |
| Partial | CFA-a | 3.21±0.04 | -5.37±2.28 | 100 | 34.7 | Over 1h |
| | CFA-p | 2.72±0.13 | -2.62±1.93 | 100 | 52.8 | Over 1h |
| Ours | CF-Nuc | 2.54±0.30 | 3.61±0.30 | 100 | 73.1 | 0.15 |
| | CF-L2 | **1.97±0.14** | **3.38±0.08** | 100 | **78.4** | **0.14** |
| Oracle | CF-SCM | *2.28±0.35* | *3.78±0.11* | *99* | *82.0* | *0.17* |

Our methods, AR-Nuc and AR-L2, offer improved scalability in high-dimensional settings. The need to consider every possible subset of the total feature set for conducting interventions in CFA-a and CFA-p results in exponential complexity relative to the total feature set. In contrast, our methods operate on learned representations of exogenous noise, achieving linear complexity with the number of neurons in the representation layer, i.e., the original feature dimension. Besides, we also examine the stability of our methods, AR-$\mathcal{L}$2 and AR-Nuc, by varying the hyper-parameters $\alpha_{mag}$ and $\alpha_{nuc}$. Results are shown in the appendix for saving space A.5.

## 5 DISCUSSION AND FUTURE WORK

To protect the vulnerable end-users toward the decision models, we enhance the feasibility of algorithmic recourse such that the users can obtain both interpretable and actionable recommendations. We achieve this by identifying and constraining the variability of the exogenous noise. Extensive experimental results have verified the effectiveness of our methods. However, one limitation remains to be addressed in future work, as our method assumes causal sufficiency with no unobserved features. One possible solution to relax such assumption is to introduce auxiliary information (e.g., instrumental variables (Wang et al., 2022) or proxy variables (Shi et al., 2020)) for more powerful identification on the structural functions. We will consider this direction in our future work.

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

# A  APPENDIX

## A.1  AGGREGATED FORMULATION OF ADDITIVE SCM

The aggregated formulation of linear additive SCMs is already present in (Shimizu, 2014), i.e., $X = AX + \sigma$, where $A$ is the linear version of $\{f_1, f_2, \cdots, f_n\}$ and $\sigma$ is the exogenous noise. In similar, we can extend such aggregation to non-linear $\{f_1, f_2, \cdots, f_n\}$ as follows:

$$\begin{pmatrix} x_1 \\ x_2 \\ \vdots \\ x_n \end{pmatrix} = \begin{pmatrix} f_1(p_a(x_1)) \\ f_2(p_a(x_2)) \\ \vdots \\ f_n(p_a(x_n)) \end{pmatrix} + \begin{pmatrix} \sigma_1 \\ \sigma_2 \\ \vdots \\ \sigma_n \end{pmatrix} = f \begin{pmatrix} p_a(x_1) \\ p_a(x_2) \\ \vdots \\ p_a(x_n) \end{pmatrix} + \begin{pmatrix} \sigma_1 \\ \sigma_2 \\ \vdots \\ \sigma_n \end{pmatrix} \tag{13}$$

Notably, the aggregated function, i.e., $f$, is defined by setting the $j$-th coordinate of $f$'s domain to be the output of $f_j$. In other words, $f \begin{pmatrix} p_a(x_1) \\ p_a(x_2) \\ \vdots \\ p_a(x_n) \end{pmatrix}_j = f_j(p_a(x_j))$. Thus, when we re-formulate the generation of data as $X = f(X) + \sigma$ with further transformations, i.e., $X = g(\sigma)$, the mutually independent $\sigma$ enables us to recover latent $\sigma$ from observational $X$. Subsequently, we can further constrain the variation of $\sigma$ and keep $f$ invariant indirectly.

---

**Algorithm 1** Illustrations of AR-Nuc and AR-$\mathcal{L}2$

---

**Require:** The collected observational dataset $\mathcal{D} = \{\mathbf{x}_i^F, \mathbf{y}_i\}_{i=1}^M$, the algorithmic recourse model (CVAE) $\mathcal{M}$, the regression model $\mathcal{M}_l = \{\psi, \phi\}$, the size of input features n, the batch size of $\mathcal{M}$ as $M_b$.

**Ensure:** The trained model $\mathcal{M}$.

 1: **Extracting the exogenous representations**:
 2: Randomly shuffle $\{\mathbf{y}_i\}_{i=1}^M$ and obtain the permuted $\widehat{\mathbf{y}}$;
 3: Construct the augmented dataset $\mathcal{D}^A = \{\mathbf{x}_i^F, \widehat{\mathbf{y}}_i\}_{i=1}^M$;
 4: Optimize the regression model $\mathcal{M}_l$ as in equation 3 by discriminating between $\mathcal{D}$ and $\mathcal{D}^A$.
 5: **Training the algorithmic recourse model**:
 6: Arrive the latent representation $\{\mathbf{z}_i\}_{i=1}^{M_b}$ through the encoder from the input $\{\mathbf{x}_i^F\}_{i=1}^{M_b}$ with target labels $\{\mathbf{y}_i'\}_{i=1}^{M_b}$;
 7: By sampling from $\{\mathbf{z}_i\}_{i=1}^{M_b}$, compute the reconstructed $\{\mathbf{x}_i^{AR}\}_{i=1}^{M_b}$ from the decoder with $\{\mathbf{y}_i'\}_{i=1}^{M_b}$;
 8: Compute the original objective $\mathcal{L}_{\text{ori}}$ of $\mathcal{M}$ with $\{\mathbf{z}_i, \mathbf{x}_i^{AR}, \mathbf{x}_i^F, \mathbf{y}_i, \mathbf{y}_i'\}_{i=1}^{M_b}$;
 9: **AR-Nuc**: Optimize the total objective as $\mathcal{L}_{\text{ori}} + \alpha_{nuc}\mathcal{L}_{\text{nuc}}$.
10: **AR-$\mathcal{L}2$**: Optimize the total objective as $\mathcal{L}_{\text{ori}} + \alpha_{mag}\mathcal{L}_{\text{mag}}$.

---

## A.2 DETAILED ALGORITHM

We put detailed illustration of our algorithm in Alg. 1.

## A.3 THEORETICAL PROOF

Throughout our appendix, we use the subscript $1 \leq j \leq n$ to index the feature, the subscript $1 \leq i \leq M$ to index the sample, and the subscript $1 \leq k \leq K$ to index the order in the conditional exponential distribution.

*Proof for Theorem 4.1.* Overall, our techniques in this proof are inspired by the previous results in (Hyvarinen et al., 2019). First, with the properties that $\sigma_{j1}$ is statistically dependent on $\mathbf{y}$, but conditionally independent of the other $\sigma_{j2}$, we have the following expression:

$$\log p(\sigma \mid \mathbf{y}) = \sum_{j=1}^n q_j(\sigma_j, \mathbf{y}). \tag{14}$$

Furthermore, based on previous well-known results (Gutmann & Hyvärinen, 2012), the universal approximation capability assumption in our theorem implies that the regression function $r$ will equal the difference of the log-densities in the two classes (namely $\mathcal{D}$ and $\mathcal{D}^A$):

$$\begin{aligned}\sum_{j=1}^n \psi_j(\phi_j(\mathbf{x}), \mathbf{y}) &= \log p(\sigma, \mathbf{u}) + \log|det\mathbf{Jg}(\mathbf{x})| - \log p(\sigma) \\ &\quad - \log p(\mathbf{y}) - \log|det\mathbf{Jg}(\mathbf{x})|,\end{aligned} \tag{15}$$

where the term $det\mathbf{Jg}(\mathbf{x})$ refers to the determinant of the Jacobian matrix of $g$, and the equality holds due to the fact that the $p(\sigma, \mathbf{y}) = p(\sigma)p(\mathbf{y})$ in $\mathcal{D}^A$. Meanwhile, based on the conditional-exponential assumption, the left side of the above equation can be simplified into the following expression:

$$\sum_j \log Q_j(\sigma_j) + \left[\sum_k \tilde{q}_{jk}(\sigma_j)\lambda_{jk}(\mathbf{y})\right] - \log Z_j(\mathbf{y}) - \log p(\sigma). \tag{16}$$

Consequently, a linear solution of $\sum_{j=1}^n \psi_j(\phi_j(\mathbf{x}), \mathbf{y})$ can be written as follows:

$$\sum_{jk} \tilde{\phi}_{jk}(\mathbf{x})v_{jk}(\mathbf{y}) + s(\mathbf{x}) + t(\mathbf{u}), \tag{17}$$

where

$$\begin{aligned}\tilde{\phi}_{jk}(\mathbf{x}) &= \tilde{q}_{jk}(\sigma_j) \\ v_{jk}(\mathbf{y}) &= \lambda_{jk}(\mathbf{y}) \\ s(\mathbf{x}) &= \sum_j \log Q_j(\sigma_j) - \log p(\sigma) \\ t(\mathbf{y}) &= \sum_j -\log Z_j(\mathbf{y}),\end{aligned} \tag{18}$$

where the representations $\tilde{\phi}(x)$ identifies exactly the $\tilde{q}(\sigma)$ in this special solution. Moreover, we show that the above solution for the regressor is the only solution up to the $\mathbf{A}, \mathbf{b}$ given in the theorem (namely, $\tilde{\phi}$ identifies $\tilde{q}$ up to a linear transformation). To this end, we collect the following equations for the points $\mathbf{y}_{l=1}^{nk+1}$ in the assumption 2 in our theorem:

$$\sum_{jk} \tilde{\phi}_{jk}(\mathbf{x}) v_{jk}(\mathbf{y}_l) + s(\mathbf{x}) + t(\mathbf{u})$$
$$= \sum_j \log Q_j(\sigma_j) + \left[\sum_k \tilde{q}_{jk}(\sigma_j) \lambda_{jk}(\mathbf{y}_l)\right] - \log Z_j(\mathbf{y}) - \log p(\sigma), \tag{19}$$

then the following matrix expression is obtained:

$$\mathbf{W}^T \tilde{\phi}(\mathbf{x}) = \mathbf{L}^T \tilde{\mathbf{q}}(\sigma) - \mathbf{z} + \mathbf{1}\left[\sum_j \log Q_j(\sigma_j) - q_0(\sigma) - a(\mathbf{x})\right], \tag{20}$$

where $\mathbf{W} \in \mathcal{R}^{nk \times (nk+1)}$ is the matrix expression of the vectors $\mathbf{W}(\mathbf{y}_l)$ $(1 \le \le nk)$, $\mathbf{L} \in \mathcal{R}^{nk \times (nk+1)}$ is the matrix form of $\lambda_{jk}(\mathbf{y}_l)$ with $j * k$ as the row index and $l$ as the column index, $\tilde{\mathbf{q}}(\sigma) \in \mathcal{R}^{nk}$ is the collection of $\tilde{q}_{jk}(\sigma_j)$, $\tilde{\phi}(\mathbf{x}) \in \mathcal{R}^{nk}$ is the representation vector, $\mathbf{z} \in \mathcal{R}^{nk+1}$ is the collections of all $t(\mathbf{y}_l) + \sum_j \log Z_j(\mathbf{y}_l)$ for different $l$, and $\mathbf{1} \in \mathcal{R}^{nk+1}$ is a vector of ones. Moreover, we subtract the first row of the above equation from its rest rows, and derive the following equation:

$$\widehat{\mathbf{W}}^T \tilde{\phi}(\mathbf{x}) = \widehat{\mathbf{L}}^T \tilde{\mathbf{q}}(\sigma) - \widehat{\mathbf{z}}, \tag{21}$$

where $\widehat{\mathbf{W}}$ and $\widehat{\mathbf{L}}$ are differences of the rows of $\mathbf{W}$ and $\mathbf{L}$ (and likewise for $\widehat{\mathbf{z}}$). Finally, since the matrix $\widehat{\mathbf{L}}$ coincides with invertible assumption (b) in our theorem, we obtain the identification results as follows:

$$\mathbf{A}\tilde{\phi}(\mathbf{x}) = \tilde{\mathbf{q}}(\sigma) - \mathbf{b}, \tag{22}$$

where $\mathbf{A} = \widehat{\mathbf{L}}^{-1}\widehat{\mathbf{W}}$ and $\mathbf{b} = \widehat{\mathbf{L}}^{-1}\widehat{\mathbf{z}}$. Notably, the unknown matrices $\mathbf{A}$ and $\mathbf{b}$ only depend on the support points $\mathbf{y}$. □

*Proof for Theorem 4.2.* First, we list the expression of $\mathbf{H}^\sigma$ and $\mathbf{H}^0$ and $\mathbf{H}$ for convenience:

$$\begin{cases} \mathbf{H}^\sigma = \{\tilde{\mathbf{q}}(\sigma_1^F) - \tilde{\mathbf{q}}(\sigma_1^{CF}), \tilde{\mathbf{q}}(\sigma_2^F) - \tilde{\mathbf{q}}(\sigma_2^{CF}), \dots, \tilde{\mathbf{q}}(\sigma_{M_b}^F) - \tilde{\mathbf{q}}(\sigma_{M_b}^{CF})\}, \\ \mathbf{H}^0 = \{\sigma_1^F - \sigma_1^{CF}, \sigma_2^F - \sigma_2^{CF}, \dots, \sigma_{M_b}^F - \sigma_{M_b}^{CF}\}, \\ \mathbf{H} = \{\phi(\mathbf{x}_1^F) - \phi(\mathbf{x}_1^{CF}), \phi(\mathbf{x}_2^F) - \phi(\mathbf{x}_2^{CF}), \dots, \phi(\mathbf{x}_{M_b}^F) - \phi(\mathbf{x}_{M_b}^{CF})\}, \end{cases}$$

where we argue that minimizing the rank of $\mathbf{H}^\sigma$ is enough to constrain the sparsity of $\mathbf{H}^0$. To be specific, assuming the statistics $\tilde{q}$ is injective, then for any feature index $j$, $\tilde{\mathbf{q}}(\sigma_2^F)_j - \tilde{\mathbf{q}}(\sigma_2^{CF})_j = 0 \Rightarrow (\sigma_2^F - \sigma_2^{CF})_j = 0$. Consequently, the number of non-zero entries of $\mathbf{H}^\sigma$ equals to that of $\mathbf{H}^0$, and the sparsity $\mathbf{H}^\sigma$ implies the sparsity of $\mathbf{H}^0$.

Based on the assumption (a) in Theorem 4.2, we can easily obtain that the Jacobian matrix $\mathrm{J}\tilde{\mathbf{q}}$ of $\tilde{\mathbf{q}}$ with respect to $\sigma$ with the size $nk \times n$ exists. Moreover, $\mathrm{J}\tilde{\mathbf{q}}$ is of full-rank, with Rank $(\mathrm{J}\tilde{\mathbf{q}}) = n$. Analogously, with the assumption (b) in Theorem 4.2, we obtain that the Jacobian $\mathrm{J}\phi$ of $\phi\mathbf{x}$ is of full-rank with Rank $(\mathrm{J}\phi) = n$ as well. Therefore, recalling the equation in Theorem 4.1 as equation 6, we have the following equation:

$$\mathbf{A}\mathrm{J}\phi = \mathrm{J}\tilde{\mathbf{q}}, \tag{23}$$

which further implies that $\mathbf{A} \in \mathcal{R}^{n \times n}$ is of full-rank with Rank$(\mathbf{A}) = n$. Therefore, as $\mathbf{A}\mathbf{H}^\sigma = \mathbf{H}$, we conclude that Rank $(\mathbf{H}^\sigma) = $ Rank$(\mathbf{H})$. Therefore, constraining the sparsity of $\mathbf{H}$ is equivalent to constraining that of $\mathbf{H}^\sigma$, which further governs the sparsity of $\mathbf{H}^0$. □

*Proof for Theorem 4.3.* First, we show that the term $\|\tilde{\mathbf{q}}(\sigma^F) - \tilde{\mathbf{q}}(\sigma^{CF})\|_2$ is bounded by $\|\phi(\mathbf{x}^F) - \phi(\mathbf{x}^{CF})\|_2$ as follows:

$$\begin{aligned} \|\tilde{\mathbf{q}}(\sigma^F) - \tilde{\mathbf{q}}(\sigma^{CF})\|_2 &= \|\mathbf{A}\phi(\mathbf{x}^F) - \phi(\mathbf{x}^{CF})\|_2 \\ &\le \|\mathbf{A}\|\|\phi(\mathbf{x}^F) - \phi(\mathbf{x}^{CF})\|_2, \end{aligned} \tag{24}$$

where the first equality is due to the results of identification, and the second inequality is due to the definition of the norm of the operator $\mathbf{A}$. Moreover, as $\mathbf{A}$ maps between finite-dimensional Hibert spaces and $\mathbf{A}$ is a continuous operator, $\mathbf{A}$ itself is bounded (e.g., $\|\mathbf{A}\| \le C$ holds). Meanwhile, recalling our assumption that $\tilde{\mathbf{q}}$ is a bi-lipschitz function, we have:

$$K_1 \|\sigma^F - \sigma^{CF}\|_2 \le \|\tilde{\mathbf{q}}(\sigma^F) - \tilde{\mathbf{q}}(\sigma^{CF})\|_2 \le K_2 \|\sigma^F - \sigma^{CF}\|_2, \qquad (25)$$

where $K_1$ and $K_2$ are Lipschitz constants. Notably, such assumpion implies that the variation of $\tilde{\mathbf{q}}$ is compactly correlated to that of $\sigma$, which is common for exponential families (Hyvarinen & Morioka, 2017). Hence, $\|\sigma^F - \sigma^{CF}\|_2$ is governed by $\|\phi(\mathbf{x}^F) - \phi(\mathbf{x}^{CF})\|_2$:

$$\|\sigma^F - \sigma^{CF}\|_2 \le \frac{1}{K_1} \|\mathbf{A}\| \|\phi(\mathbf{x}^F) - \phi(\mathbf{x}^{CF})\|_2, \qquad (26)$$

where minimizing $\|\phi(\mathbf{x}^F) - \phi(\mathbf{x}^{CF})\|_2$ is enough to constrain $\|\sigma^F - \sigma^{CF}\|_2$. $\qquad\square$

## A.4 Experimental Details

**Details on Implementation of baselines**  The CFVAE baseline serves as the underlying algorithmic recourse model for our AR-Nuc and AR-$\mathcal{L}2$ methods. We follow the implementations in (Mahajan et al., 2019), using Multi-layer-perception (MLP) layers to estimate $\mu_{y'}$ and $\sigma^2_{y'}$ in the encoder branches. The black box ML model $h$ is also an MLP classifier. We use the Adam optimizer (Bock et al., 2018) with an initial learning rate of $0.01$ for $h$ and $\mathcal{M}$. The batch size $M_b$ is set to 64 in all our experiments. The original implementations of AR-SCM and CFVAE in Pytorch by (Mahajan et al., 2019) are publicly available[3]. The CEM[4] and CFA[5] methods are also open-source on GitHub.

**Details on Implementation of our models**  First, we detail the architecture of the underlying algorithmic recourse model, namely CFVAE, as we implemented in our paper. The branch of the encoder for estimating $\mu_{y'}$ contains 5 Multi-layer-perception (MLP) layers with $Elu$ as the activation functions and the batch-normalization after each layer, with the same architecture of the branch for estimating $\sigma^2_{y'}$, except for a $Sigmoid$ function after the final layer to constrain the variance. Meanwhile, the decoder architecture follows the above protocols as well. The training loss is set to the BCE loss, as the domain label is binary. Besides, the black box ML model $h$ consists of two MLP layers with the $Elu$ function. Notably, to control the effect from redundant variables, the architecture and optimization introduced above keep exactly the same for our methods (AR-Nuc and AR-$\mathcal{L}2$) and the CFVAE (Mahajan et al., 2019). Regarding AR-Nuc and AR-$\mathcal{L}2$, for the regression system, we use a neural network with three MLP layers to model $\phi : \mathbb{R}^n \mapsto \mathbb{R}^n$, where the number of units in the hidden layers is 2n, except for the final layer with n units. The nonlinearity used is ReLU. Each $\psi_j : \mathbb{R}^2 \mapsto \mathbb{R}$ is modeled with three MLP layers and ReLU functions, for $1 \le j \le n$. The regression system is trained using the Adam optimizer (Bock et al., 2018) with an initial learning rate of $0.001$. The margin $\beta$, and the hyperparameters $\alpha_n$ and $\alpha_m$ for controlling AR-Nuc and AR-$\mathcal{L}2$, are set to $0.2$, $2$, and $2$ respectively, in all our experiments.

**Details on Dataset**  We then detail the simulation on the German Loan dataset, with the same protocols in (Kanamori et al., 2021) as follows:

$$
\begin{aligned}
G &: U_G, & U_G &\sim \mathrm{Bernoulli}(0.5) \\
A &:= -35 + U_A, & U_A &\sim \mathrm{Gamma}(10, 3.5) \\
E &:= -0.5 + \left(1 + e^{-\left(-1 + 0.5G + \left(1 + e^{-0.1A}\right)^{-1} + U_E\right)}\right)^{-1}, & U_E &\sim \mathcal{N}(0, 0.25) \\
L &:= 1 + 0.01(A - 5)(5 - A) + G + U_L, & U_L &\sim \mathcal{N}(0, 4) \\
D &:= -1 + 0.1A + 2G + L + U_D, & U_D &\sim \mathcal{N}(0, 9) \\
I &:= -4 + 0.1(A + 35) + 2G + GE + U_I, & U_I &\sim \mathcal{N}(0, 4) \\
S &:= -4 + 1.5\mathbb{I}_{\{I > 0\}}I + U_S, & U_S &\sim \mathcal{N}(0, 25).
\end{aligned}
\tag{27}
$$

---

[3] https://github.com/divyat09/AR-feasibility
[4] https://github.com/IBM/Contrastive-Explanation-Method
[5] https://github.com/amirhk/recourse

Meanwhile, we generate the class label $\mathbf{y}$ following (Karimi et al., 2020):

$$\mathbf{y} \sim \text{Bernoulli}\left(\left(1 + e^{-0.3(-L-D+I+S+IS)}\right)^{-1}\right). \tag{28}$$

Besides, we provide details on the sample number for each dataset. For the synthetic dataset and semi-synthetic German Load dataset, we set $M = 10000$ as the number of samples. For the real-world Diabetes dataset, we have $M = 768$ samples. Such variation on the samples size also verifies that our methods does not rely on huge data samples. To report the out-of-sample prediction results, we randomly split the each dataset into the training/testing domains with ratio as $0.7/0.3$.

**Details on generating the high-dimensional Dataset**    We have augmented our research by incorporating additional experiments involving a synthetic dataset in our study. The synthetic dataset is designed with a feature dimension of 80. The rationale behind employing synthetic data is twofold: (a) most widely used, realistic datasets possess relatively small feature dimensions; (b) real-world data lacks an underlying structural causal model (SCM), rendering it infeasible to verify whether generated explanations align with the SCM model. Specifically, we extend the synthetic setting in our paper to encompass a high-dimensional scenario with 80 dimensions:

$$
\begin{aligned}
\mathbf{x}_1 &\sim N\left(\mu_1, \sigma_1\right); \\
\mathbf{x}_2 &\sim N\left(\mu_2, \sigma_2\right); \\
\mathbf{x}_3 &\sim N\left(\mu_1, \sigma_1\right); \\
\mathbf{x}_4 &\sim N\left(\mu_2, \sigma_2\right); \\
&\cdots, \\
\mathbf{x}_8 &\sim N\left(\mu_2, \sigma_2\right); \\
\mathbf{x}_9 &\sim N\left(k_1 * \left(\mathbf{x}_1 + \mathbf{x}_2 + \mathbf{x}_3 + \mathbf{x}_4\right)^2 + b_1, \sigma_3\right); \\
\mathbf{x}_{10} &\sim N\left(k_1 * \left(\mathbf{x}_5 + \mathbf{x}_6 + \mathbf{x}_7 + \mathbf{x}_8\right)^2 + b_1, \sigma_3\right).
\end{aligned}
\tag{29}
$$

In order to augment the original dataset $X^F$ and create a more complex structure, additional variables $\mathbf{x}_{i*10+1}$ to $\mathbf{x}_{i*10+10}$ are generated for $1 \leq i \leq 7$, following the same procedure as $i = 0$. Subsequently, a random permutation is applied to shuffle the variables $\mathbf{x}1$ to $\mathbf{x}80$. This permutation aims to challenge the preservation of the original structure in $X^F$.

To ensure the integrity of the modified dataset, a feasibility check is performed. Specifically, for each sample, the following conditions are validated: - If $\mathbf{x}_i, \mathbf{x}_{i+1}, \mathbf{x}_{i+2}, \mathbf{x}_{i+3}$ increase, then $\mathbf{x}_{i+9}$ must also increase for $i = 10k + 1$, where $0 \leq k \leq 7$. - If $\mathbf{x}_i, \mathbf{x}_{i+1}, \mathbf{x}_{i+2}, \mathbf{x}_{i+3}$ decrease, then $\mathbf{x}_{i+9}$ must also decrease for $i = 10k + 1$, where $0 \leq k \leq 7$. - Similarly, for each sample, if $\mathbf{x}_i, \mathbf{x}_{i+1}, \mathbf{x}_{i+2}, \mathbf{x}_{i+3}$ increase, $\mathbf{x}_{i+5}$ should also increase, and if they decrease, $\mathbf{x}_{i+5}$ should also decrease, for $i = 10k + 5$, where $0 \leq k \leq 7$.

**Details on the Regression System**    Moreover, we perform extra experiments to illustrate the behaviour of our regression system for extracting the exogenous representations. To be specific, we report the training process on the Diabetes dataset in Figure 5, where the convergent training loss indicates that the model indeed achieves nearly the universal approximation capability (which is critical for identifying the exogenous noise in our theorem).

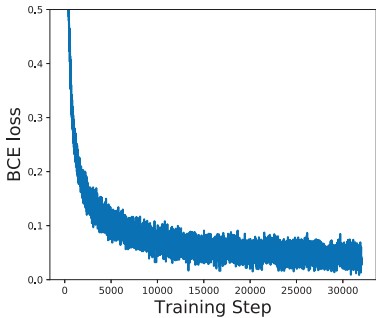

**Figure 5: The training curve of our regression system.**

## A.5 STABILITY ANALYSIS

Finally, we have tested the stability of our methods, AR-$\mathcal{L}2$ and AR-Nuc, by varying the hyper-parameters $\alpha_{mag}$ and $\alpha_{nuc}$. The in-sample prediction results in Table 6 show that (a) our methods have weak effects on the feasibility when $\alpha \leq 0.1$; (b) our AR-$\mathcal{L}2$ and AR-Nuc does not ruin other metrics such as proximity when improving the feasibility; (c) the feasibility achieved by our methods does not rely on the sophisticated tuning of hyper-parameters $\alpha_{mag}$ and $\alpha_{nuc}$ (only require the hyper-parameter not to be too small).

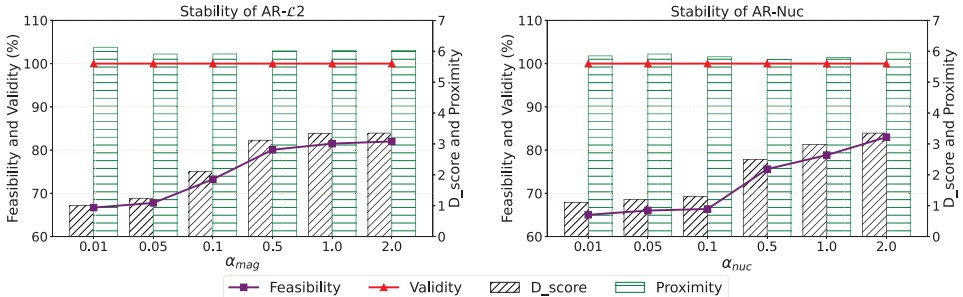

**Figure 6: Stability of of our methods.**

