# OpenReview forum: "Feasible Algorithmic Recourse Without Explicit Structure Prior"
_ICLR.cc/2024/Conference — Submitted to ICLR 2024_

### Official Review · Reviewer_VUqJ · 2023-10-27

**Soundness:** 4 excellent
**Presentation:** 2 fair
**Contribution:** 4 excellent
**Rating:** 8
**Confidence:** 4

**Summary:**

In this paper, the authors proposed a method for algorithmic recourse that reflects causal structures even when the causal structure is unknown.
Specifically, they assume a nonlinear structural equation model as the causal structure and adjust only the exogenous variables of this structural equation model in the context of algorithmic recourse.
To faithfully control the exogenous variables for an unknown structural equation model, the authors proposed using the nonlinear ICA from (Hyvarinen et al., 2019).
In their nonlinear ICA, under certain assumptions, the difference between exogenous variables before and after manipulation can be estimated up to linear transformations.
The authors employed the degrees of freedom of the difference between exogenous variables before and after manipulation (i.e., the solution of nonlinear ICA) as a penalty in algorithmic recourse, using either the nuclear norm or L2 norm.
This penalty restricts the exogenous variables to be manipulated only within the range that appropriately reflect the causal structure.
Finally, the authors incorporate the above norm as a penalty during the training of a Conditional VAE (CVAE) that generates algorithmic recourse.
The authors demonstrated through experiments using both synthetic and real-world data that the CVAE trained with the proposed method indeed realizes algorithmic recourse that is faithful to the causal structure.

**Strengths:**

The strength of this paper is the development of algorithmic recourse that is faithful to the causal structure even when the causal structure is unknown.

**Originality, Quality**

The originality of this paper stems from the use of nonlinear ICA of (Hyvarinen et al., 2019).
It enables algorithmic recourse that is faithful to the causal structure without the need of estimating it.
The authors proposed the use of the solution of nonlinear ICA as a penalty term, leveraging the fact that it corresponds to the difference between exogenous variables before and after manipulation, up to linear transformations.
If one wishes to recover the causal structure, one needs to estimate the linear transformation part.
However, for the faithful algorithmic recourse, the authors showed that it is not necessary, thus avoiding the estimatin of the causal structure.
This concept is a significant contribution of this research.

**Clarity**

Please refer to "Weakness" below.

**Significance**

The significance of this research is the development of algorithmic recourse that is faithful to the causal structure even when the causal structure is unknown.
As mentioned above, the fact that there is no need to estimate the causal structure itself would be an essential contribution of this research.

**Weaknesses:**

The weakness of this paper is its lack of clarity.
This research is based on nonlinear ICA of (Hyvarinen et al., 2019).
However, the paper only discusses the procedure and theoretical properties of this nonlinear ICA, omitting all fundamental details regarding why the method can solve nonlinear ICA.
Consequently, for readers who are not familiar with the nonlinear ICA of (Hyvarinen et al., 2019), it becomes challenging to understand fundamental contributions of this research, such as the relationship between nonlinear ICA and nonlinear structural equation modeling and the critical role this relationship plays in this study.
While the details of nonlinear ICA procedures and its theoretical properties are undoubtedly important, it is equally essential to convey the overall picture of the proposed method (e.g., the use of exogenous variable manipulation range in CVAE penalty) to readers.
Therefore, I would like to recommended the authors to reconsider the structures of Sections 2 and 3 to effectively communicate these aspects to the readers.

**Questions:**

* Please reconsider the structures of Sections 2 and 3 so that the main contribution of the paper to be clear to the readers not familiar with nonlinear ICA of (Hyvarinen et al., 2019).

---

### Official Review · Reviewer_Nx58 · 2023-10-29

**Soundness:** 2 fair
**Presentation:** 2 fair
**Contribution:** 3 good
**Rating:** 5
**Confidence:** 2

**Summary:**

This paper approaches the feasibility issue of CEs (where changes to the input instance are not actionable) from a new angle, and without access to an explicit causal structure. The paper introduces CF-Nuc and CF-L2, which work by identifying and constraining the variability of exogenous noise instead of directly manipulating features, in order to preserve causal relationships in the data. The authors validate their methods on synthetic, semi-synthetic, and real-world datasets, showing improved feasibility compared to baseline models that have prior knowledge of the causal graph.

**Strengths:**

- The approach to generating feasible CEs, focusing on constraining the variation of the exogenous noise in order to maintain causal relationships, is relatively original, and could be seen as a valuable contribution to the literature.
- The authors conduct well structured experiments to demonstrate the workings and effectiveness of their method. The transition from synthetic through to real world data is useful.
- The method's potential to generate more feasible CEs could have practical implications in various fields where interpretability of ML models is crucial.
- The authors test the method's effectiveness in higher-dimensional settings.
- Sufficient details are provided to allow reproducibility.

**Weaknesses:**

- While the paper makes decent advancements, the main limitation of the assumption that all features for causal sufficiency are observed likely does not hold in real world scenarios. Further discussion on this point would be useful as this is a fairly strong practical limitation.
- I fear the paper may be challenging to access for readers who are not deeply familiar with the literature on these topics. I had to spend a fair amount of time reading this paper to familiarize myself with the main concepts.

**Questions:**

1. The approach assumes causal sufficiency, which can be a strong assumption in many real-world scenarios where hidden confounders and unobserved variables can significantly affect the outcomes. Could the authors comment on potential extensions of their method to handle such scenarios? How such challenges would be handled is not obvious.

---

### Official Review · Reviewer_u6SK · 2023-10-31

**Soundness:** 2 fair
**Presentation:** 1 poor
**Contribution:** 2 fair
**Rating:** 3
**Confidence:** 3

**Summary:**

The authors propose that the process of algorithmic recourse on input features can be seen as the manipulation of exogenous noise in each sample, while preserving the structural causal relationships among features. To implement this concept, they indirectly preserve causal relationships by controlling the variation of exogenous noise using non-linear ICA. They claim that the variation of exogenous noise is influenced by the representations learned by the exogenous regressor. In practice, they propose two methods, AR-L2 and AR-Nuc, which respectively control the magnitude and sparsity of variations in exogenous representations. In these two methods, the regularization terms measured by either the 2-norm or the nuclear norm of the value matrix is added to the objective function of CFVAE. The numerical results show that the proposed method has better D-score.

**Strengths:**

- The paper provides some nice ideas to integrate nonlinear ICA into the recourse generation problem. The paper shows that there is a link (though very weak) between the properties of the matrix $H$ and the sparsity of the exogenous variables.

**Weaknesses:**

I have some concerns regarding the nuclear and 2-norm regularization terms:
1. On page 5, the authors claim ``To this end, we restrict the variation vector of the exogenous noise to be sparse by minimizing the rank of the matrix $H$.”. This seems to be wrong. A matrix $H$ can be low rank but dense.
2. $\|A\|$ can be so large that minimizing $\|H\|$ may not lead to a smaller 2-norm of the $\sigma$.
Thus, there is no theoretical guarantee that incorporating $H$ in the training objective function can lead to a better $\mathcal M$.

There is little theoretical contribution in this paper: Theorem 3.1 is straightforward from Hyvarinen et al. (2019), while Theorem 3.2 and 3.3 are simple results from linear algebra. After reading through this paper several times, I feel that this paper is simply combining Hyvarinen et al. (2019) and Mahajan et al. (2019) in order to have a new loss function. Thus, I see limited merit in this paper.


The paper is poorly written. Many terms are not introduced, below is a partial list:
1. $x^{AR}$ is not defined in Eq. (7)
2. $M_b$ is not defined in Eq. (8)
3. Typo: Monte-Carlo between Eq. (10) and (11). Also, $\mathcal M$ is not defined.
4. Algorithm 1 is confusing: $\mathcal M$ is the input and also is the output. $\mathcal M_l$ is the input, but $\mathcal M_l$ is retrained in line 4 of the algorithm.
5. Equation (24): bracket missing on the first line.

**Questions:**

1. Why should we have the term $\mathcal L_{recon}$ in the objective function of (11)? If we add $\mathcal L_{nuc}$ or $\mathcal L_{mag}$ in the objective function, then $\mathcal L_{recon}$ becomes redundant, am I right? What are the interplay between $\mathcal L_{recon}$ iand $\mathcal L_{nuc}$ or $\mathcal L_{mag}$ in this problem?

2. I am confused about the contributions of the paper. I hope that the authors can summarize a few key points to strengthen their contributions (what is new in this paper that cannot be found anywhere else? what is the new tool and technique that is used in this paper? etc.)

---

### Official Review · Reviewer_sk1y · 2023-10-31

**Soundness:** 2 fair
**Presentation:** 1 poor
**Contribution:** 2 fair
**Rating:** 3
**Confidence:** 3

**Summary:**

While previous works have incorporated causality into algorithmic recourse to capture real-world constraints, they rely on inaccessible prior Structural Causal Models (SCMs) or complete causal graphs. To maintain the causal relationships without such explicit prior causal knowledge, the authors suggest a formulation that exploits the relation between feature perturbations and exogenous noise perturbations using non-linear Independent Component Analysis (ICA). The authors develop a regularization term, that when added to the CFVAE objective, results in more causally aligned counterfactual explanations.

**Strengths:**

**New recourse method to generate recourse adhering to causal constraints:** The authors use the connection between SCMs and linear structural equation models to generate recourses that adhere to the linear SCM’s causal constraints. For the author’s suggested method, the exact SCM can be unknown, but need to be hypothesized to be linear.

**Competitive performance**: Relative to the compared methods, the suggested method does perform significantly better than its competitors. However, this comparison might be misleading which is further discussed among the weaknesses.

**Weaknesses:**

**Contribution**: What the authors suggest is similar to “Backtracking Counterfactuals” [1] where it is suggested tracing back counterfactuals such that the causal laws (i.e., structural equations) remain unchanged in the counterfactual world while recourses are induced by altering the initial conditions (i.e., changing exogenous variables and propagating them through the SCM). The authors write that “[t]he lack of prior causal knowledge makes direct identification or approximation of these functions impossible. To overcome this, we propose that the process of the algorithmic recourse on the input features can be modelled as solely the manipulation of the exogenous noise of each sample, while the structural causal relationships among features remain.” Conceptually, this is similar to backtracking, but the authors fail to discuss how [1] relates to their work, and most importantly how it is different from [1].

**Confusing notation and lack of clarity**: The paper fails to clarify their notation and methodologies, making it difficult for readers to follow the argument as the notation is not clear or inconsistent. Also, the paper tends to jump from one step to another without providing much explanation (e.g., see equation 2).  Some examples regarding these points include:
  - Sometimes vectors are bold, sometimes they are not. Sometimes vectors are capitalized and non-bold.
  - I cannot follow how the left-hand side of equation 2 implies the right-hand side. Can the authors clarify this?
  - Is the outcome y multidimensional (as it is bold in your notation)?
  - “The exogenous noise σ is conditionally exponential of order K of y.” Is this a fixed term? I never have heard or read this. Can the authors clarify what that means?
  - It’s my understanding that you are using additive SCMs; why do you then use a general function f to describe the SCM. Wouldn’t a linear function in form of a standard adjacency matrix suffice that excludes recursions?
  - It remains unclear to me what the connection between being able to identify noise and obtaining sparse and causally correct recourses is. It is unclear how adding the proposed regularizes would lead to causally correct counterfactuals. Can the authors please elaborate?

**Empirical evaluation**: The experimental evaluation appears unfair and might therefore be misleading. For example, the paper compares its results to CEM, which does no use SCMs, and to CFVAE, which estimates structural equations, while your method takes the (linear) SCM implicitly as given by using ICA. Could this be the reason why your method outperforms since all the experiments rely on data that is generated by linear structural equation models? It would thus probably be fairer to compare your method to standard causal recourse methods such as MINT and extend the experimental results to cases where data comes from other (generated) distributions that goes beyond linear SCMs. Finally, another reasonable work to compare is [2].

----
**References**

[1] Von Kügelgen et al (2023), “Backtracking Counterfactuals”, 2nd Conference on Causal Learning and Reasoning, 2023 (CLEAR)

[2] Pawelczyk et al (2022), “Decomposing Counterfactual Explanations for Consequential Decision Making”, Workshop on Socially Responsible Machine Learning (SRML) @ ICLR 2022

**Questions:**

- I don’t quite get the identification of exogenous noise part. Why do you randomly permute the labels? Is this your contribution, or is this Hyvarinen et al’s algorithm?
- Doesn’t f need some explicit constrains to entail realistic counterfactual explanations?
- How is x^R = g(\sigma^R). When f is linear, I see this. When it is not, it’s not immediate.

---

### Official Review · Reviewer_k9bp · 2023-11-01

**Soundness:** 3 good
**Presentation:** 3 good
**Contribution:** 3 good
**Rating:** 6
**Confidence:** 3

**Summary:**

The paper proposes a novel formulation for causally-aware recourse generation, treating counterfactual manipulations as shifts in the exogenous noise. Based on identification results for non-linear Independent Component Analysis (ICA), the paper proposes two methods in generating algorithmic recourses while preserving causal relationships between features and demonstrate their effectiveness on synthetic, semi-synthetic, and real-world data.

**Strengths:**

- The formulation is novel and potentially impactful.
- Claims for the exogenous regressor and constraints are theoretically grounded.
- The paper substantially demonstrated the effectiveness of methods in feasibility and validity.

**Weaknesses:**

- While the paper claims to provide feasible, actionable, and interpretable algorithmic recourse, I have not seen substantial discussion or evaluation on actionability and interpretability.
- The paper is not exactly easy to follow. Some intuitive examples comparing resulted recourses with one generated by other methods in a real-world scenario can help with the readability.
- There is a typo "monto-carlo" near equation 11.

**Questions:**

- I hope the author can address the concerns above.

---

### Meta-Review · Area_Chair_KYJf · 2023-12-12

**Metareview:**

This paper outlines a method to find counterfactual explanations that obey causal constraints. The proposed method seeks to preserve causal relationships by controlling the variation of exogenous noise through ICA. The paper uses this idea to develop two methods -- AR-L2 and AR-Nuc -- that can promote alignment with causal constraints.

*Strengths*

- Scope: The paper studies an important problem in algorithmic recourse -- i.e., how to ensure feasibility with respect to unknown causal structure.

- Originality: The proposed method -- which uses exogenous noise to maintain causal relationships -- highlights a different avenue to make use of information in this task. The idea is original and could be used to develop a wide range of methods in this domain.

*Weakenesses*

- Assumptions: The proposed method requires that the underlying SCM is linear -- this may be a reasonable assumption but should be studied in greater detail.

- Soundness: The proposed method cannot handle actionability constraints. This issue is neither disclosed nor studied. The method may output CEs that appear feasible but violate actionability constraints.

- Clarity: Multiple reviewers noted the lack of clarity. This is a minor point that could have been addressed through a potential rebuttal.

**Justification For Why Not Higher Score:**

The authors failed to address key weaknesses identified by the reviewers.

**Justification For Why Not Lower Score:**

N/A

---

### Decision · Program_Chairs · 2024-01-16

Reject